# Preparation and Performance of a Novel ZnO/TM/PET Composite Negative Ion Functional Fiber

**DOI:** 10.3390/polym16101439

**Published:** 2024-05-19

**Authors:** Mengxin Zhang, Jishu Zhang, Xin Lu, Jianbing Wu, Jiajia Peng, Wei Wang, Jin Tao

**Affiliations:** 1School of Textile Garment & Design, Changshu Institute of Technology, Changshu 215500, China; 20215215127@stu.suda.edu.cn (M.Z.); 201800012@cslg.edu.cn (J.W.); 202000031@cslg.edu.cn (J.P.); wwang8860@clsg.edu.cn (W.W.); taoj@cslg.edu.cn (J.T.); 2College of Textile and Clothing, Suzhou University, Suzhou 215031, China

**Keywords:** tourmaline, ZnO/TM composite, negative ion functional fiber

## Abstract

Using zinc oxide (ZnO), tourmaline (TM), and polyethylene terephthalate (PET) as main raw materials, a novel ZnO/TM/PET negative ion functional fiber was created. The rheological properties of a ZnO/TM/PET masterbatch were investigated; the morphology, XRD, and FT-IR of the fibers were observed; and the mechanical properties, thermal properties, and negative ion release properties of the new fiber were tested. The results showed that the average particle size of the ZnO/TM composite is nearly 365 nm, with an increase in negative ion emission efficiency by nearly 50% compared to the original TM. The apparent viscosity of fiber masterbatch decreases with the increase in the addition of the ZnO/TM composite, and the rheological properties of the PET fiber masterbatch are not significantly effected, still showing shear thinning characteristics when the amount of addition reaches 10%. The ZnO/TM composite disperses well in the interior and surface of the ZnO/TM/PET fiber matrix. The prepared ZnO/TM/PET fiber has excellent properties, such as fineness of 1.54 dtex, glass transition temperature of 122.4 °C, fracture strength of 3.31 cN/dtex, and negative ion release of 1640/cm^3^, which shows great industrialization potential.

## 1. Introduction

Negative air ions (NAI) exert unique effects on the human body, such as sterilization, air freshening, and functional balance regulation, and are acclaimed as “vitamins and growth factors” in the air [1,2,3]. In an environmental evaluation system, the NAI concentration is a crucial indicator for assessing air quality [4]. However, in recent years, with the excessive exploitation and irrational utilization of natural plant resources by human beings, the NAI concentration around our bodies is getting lower and lower [5], which poses a significant threat to human health. Therefore, the development of a functional fabric (especially fabric that one can wear close to the skin) that can release negative air ions to provide more shelter for the human living environment, as shown in Figure 1, has become an important topic of research in the textile industry [6,7,8].

At present, there are three main methods for the preparation of negative air ion textiles: (1) surface coating [9,10], (2) blending [11,12,13], and (3) copolymerization [14]. For example, Jishu Zhang et al. used an ultra-fine tourmaline powder to finish their knitted cotton fabric and developed a negative ion functional knitted fabric [9]. Xiaowen Dong et al. used the silica sol-gel method to make negative ion functional cotton socks by double-dipping and double-rolling, and the metabolism and reproduction of bacterial colonies on the skin could be suppressed effectively while wearing it [10]. Xinglei Zhao et al. prepared polyvinylidene fluoride (PVDF) fibers containing negative ion powder (NIP) via electrostatic spinning to serve as air purification materials [11]. Yuanchi Zhang et al. used graphene oxide to modify a polyurethane/tourmaline nanocomposite to prepare the fibers [12]. Zhi Chen et al. prepared a PET/Ge composite fiber with 1–3% mass fraction of inorganic germanium powder through the melt-composite-spinning method [13]. Kaijun Zhang et al. prepared negative ion polyester slices through the copolymerization method, and then prepared functional anion fibers through the melt-spinning method [14]. These studies and the development of these fibers greatly expand the embodiment of negative ion functions in the textile field and provide important references for the subsequent development of higher performance negative ion function textiles.

Tourmaline is a common additive for the development of negative ion functional fibers. Its special crystal structure causes it to produce piezoelectric and pyroelectric effects under the external temperature, pressure change, or spontaneous polarization effect, so that its surface generates charges and then ionizes air molecules to form negative ions [15,16,17,18]. In recent years, scholars have conducted a series of studies on the factors affecting the negative ion emission efficiency of tourmaline, mainly through doping modified tourmaline [19,20,21] and small particle size tourmaline [22,23,24] to obtain much higher negative ion release. For example, Chaohui Wang et al. prepared a high whiteness tourmaline composite powder through the coating method with titanium dioxide (TiO_2_) as the coating agent and stearic acid as the coupling agent. When the mass fraction of tourmaline is 11%, the whiteness of the coating is 87.1%, and the increment of negative ion release exceeds 800 cm^−3^ [19]. Chaohui Wang et al. prepared graphene/tourmaline composites through the ball milling method. This method can make the negative ion release performance of the composite material more than 11.9% higher than that of tourmaline [20]. Kui Cui et al. used silane coupling agent KH-550 as a modifier of tourmaline to introduce polymerable carbon–carbon double bonds to the surface of tourmaline, thus significantly increasing its negative ion release and far-infrared emissivity [21]. Huihui Ding et al. prepared ultra-fine tourmaline powder by grinding for a textile finishing application, and the test results showed that the negative ion emission increased by nearly 80% [22]. Yingmo Hu et al. successfully improved the activation index of tourmaline powder and the dispersion of modified tourmaline by reacting it with 3% tourmaline in a specific environment (toluene environment at 60 °C) [23]. Guorui Huang et al. prepared a nano-tourmaline surface treatment agent through the sanding process, the minimum average particle size of which dispersion is 44 nm, and the negative ion release amount of the wall cloth after treatment reaches 6500 cm^−3^ when the mass fraction of dispersant added is 30% [24]. These studies provide ideas and references for the development of this subject.

In this paper, selecting tourmaline as the negative ion additive and combining it with a ZnO nano photocatalyst, a novel ZnO/TM composite material and a novel ZnO/TM/PET negative ion functional fiber were prepared. The rheological properties of the ZnO/TM/PET composite and the morphology, mechanical properties, thermal properties, and anion release properties of the ZnO/TM/PET fiber were studied to provide theoretical and experimental references for developing higher performance negative ion functional fibers.

## 2. Materials and Methods

### 2.1. Materials

The main experimental materials used in this study were zinc oxide (Shanghai Macklin Biochemical Co., Ltd., Shanghai, China, 99.9% metals basis, 200 nm), tourmaline (Shijiazhuang Bo Rui Building Materials Co., Ltd., Shijiazhuang, China), ethanol anhydrous (Shanghai Macklin Biochemical Co., Ltd., pharmaceutical grade, 99.5%), sodium polyacrylate (Shanghai Macklin Biochemical Co., Ltd., 50% water solution), silane A174 (Shanghai Macklin Biochemical Co., Ltd., 97%), a polyester chip (Shanghai Thousand New Materials Co., Ltd., Shanghai, China) and polyester fiber spinning oil (Tianjin University of Technology textile auxiliary Co., Ltd., Tianjin, China).

### 2.2. Equipment

A planetary ball mill (Shenzhen Jitong Technology Development Co., Ltd., Shenzhen, China, PBM-1), electronic analytical balance (Mettler Toledo Instruments Co., Ltd., Shanghai, China, PL203), vacuum drying oven (Shanghai Jinghong Experimental Equipment Co., Ltd., Shanghai, China, DZF), twin screw melt spinning machine (Changzhou Ling Fiber Textile Machinery Co., Ltd., Changzhou, China, JSH-20), twin screw extruder (Nanjing Kangfa Rubber Machinery Manufacturing Co., Ltd., Nanjing, China), muffle furnace (Changzhou Xinfang Testing Equipment Co., Ltd., Changzhou, China, SX2-5-12) and magnetic stirrer (Changzhou Ronghua Instrument Co., Ltd., Changzhou, China, HJ-6A) were used.

### 2.3. Sample Preparation

#### 2.3.1. Preparation of ZnO/TM Composite Powder

ZnO and TM were weighed according to the mass ratio of 1:2 and mixed with 10 mL of anhydrous ethanol as a wet grinding solvent. This mixture was placed in a stainless-steel ball grinding tank (volume of 100 mL), where we set the ball mass ratio of 1:3, ball mill speed of 300 RPM, and rotation direction of positive and negative rotation for 5 min, and then ground it for 15 h to obtain the ZnO/TM composite slurry.

#### 2.3.2. Preparation of Functional Polyester Cuttings

The ZnO/TM composite slurry was continuously magnetically stirred for 2 h with KH-570 as modifier and PAAS as dispersant at the ratio of 18.5:0.9:1.9:18.7 (wt%). Polyester cuttings were dried at 120 °C for 24 h in a vacuum drying oven, and then were mixed evenly with the functional slurry in a high-speed stirrer. The mixture was extruded in a twin-screw extruder at 240 °C for the first time to produce functional pellets. After drying at 120 °C for 24 h in a vacuum drying oven, the prepared functional pellets were extruded for a second time to obtain a functional masterbatch.

#### 2.3.3. Preparation of ZnO/TM/PET Negative Ion Functional Fiber

The functional pellets and polyester cuttings were evenly mixed in a high-speed stirrer and then dried for 48 h at 120 °C in a vacuum drying oven. Eight novel ZnO/TM/PET fiber samples were prepared by melt spinning mechanism. The process flow chart is shown in Figure 2. The temperature in each working area of the spinning equipment is shown in Table 1, and the process parameters are shown in Table 2. The eight kinds of ZnO/TM/PET negative ion functional fiber samples were numbered O_1_, O_2_, A_1_, A_2_, B_1_, B_2_, C_1_, and C_2_, and the raw material formula and corresponding precursor multiple of each sample are shown in Table 3. 

### 2.4. Testing Method

#### 2.4.1. Particle Size Testing

The size and distribution of the particles were analyzed before and after composite preparation using a zeta potential and particle size analyzer (90 Plus Zeta, Brookhaven Instruments Corporation, Holtsville, NY, USA).

#### 2.4.2. Rheological Performance Testing

The effect of shear rate on the apparent viscosity of the ZnO/TM/PET functional masterbatch with different addition amounts was tested by a rotating rheometer (AR2000, TA Instruments, New Castle, DE, USA). The test temperature is 270 °C, and the shear rate ranges from 1 to 3000 s^−1^.

#### 2.4.3. SEM

Scanning electron microscopy (ΣIGMA, ZEISS, Oberkochen, Germany) was used to observe the longitudinal morphology and cross-sectional morphology of ZnO/TM/PET fiber samples to analyze the dispersion of the ZnO/TM composite powder in the fibers. The samples were processed by gold sputtering with an accelerating voltage of 20 kV.

#### 2.4.4. Measurement of Fiber Fineness

The fibers were cut into a certain length sample, denoted L (m), then wetted at standard atmospheric pressure for 16 h. Afterwards, their weight was tested with an electronic analytical balance and denoted X (g). The calculation formula of fiber fineness T (dtex) is shown in Equation (1).
T(dtex) = (X/L) × 10^4^(1)

#### 2.4.5. XRD

X-ray diffraction (D/max-2200/PC, Rigaku, Tokyo, Japan) was used to test the phase structure of powder and the crystallinity of fibers, with Cu target, Kα radiation, λ = 0.15406 nm, tube voltage of 40 kV, tube current of 40 mA, and a scanning range of 10° to 80°.

#### 2.4.6. FTIR

Fourier transform infrared spectroscopy (Nicolet iS10, Thermo Fisher Scientific, Waltham, MA, USA) was used to analyze the functional groups of the fibers. The fiber samples were pressed into a KBr pellet at a pressure of 20 KN, and the scanning range is 4000 to 400 cm^−1^.

#### 2.4.7. Negative Ion Release Performance Testing

A negative ion detector (DR407M, Wenzhou Darong Textile Instrument Co., Ltd., Wenzhou, China) was used to test the negative ion release performance of fiber samples. The reference standards were “Detection and Evaluation of the amount of negative ions in Textiles” (GB/T 30128-2013 [25]) and “Inspection methods for Import and Export Functional Textiles Part II: Negative Ion Content” (SN/T 2558.2-2011 [26]).

#### 2.4.8. Thermal Performance Testing

Differential scanning calorimeter (STA 449 F3, NETZSCH Scientific Instruments Trading Ltd., Selb, Germany) was used to test the thermal performance of fiber samples under a nitrogen atmosphere, with a heating rate of 10 °C/min and temperature range from 10 °C to 360 °C, and the temperature curve was recorded.

A thermogravimetric analyzer (model TGA/SDTA 851, Mettler Toledo Instruments Ltd.) was used to test the thermal performance of fiber samples under a nitrogen atmosphere, with a heating rate of 10 °C/min and temperature range from 25 °C to 800 °C, and the weight loss curve was recorded.

#### 2.4.9. Heat Shrinkability Testing

The samples of 60 cm–70 cm in length were put flat on the velvet plate. After holding it at the prescribed tension for 30 s under standard atmospheric conditions, the length of the filament was determined to be L_1_. The filament was completely immersed in a mesh bag and boiled at this temperature for 10 min. The length of the filament measured at standard atmospheric conditions was L_2_ when the specified tension was applied again. The calculation formula of the fiber thermal shrinkage rate S (%) is shown in Equation (2).
S = (L_1_ − L_2_)/L_1_ × 100%(2)

#### 2.4.10. Mechanics Performance Testing

An electronic single fiber strength tester (YG006, Ningbo Textile Instrument Factory, Ningbo, China) was used to test the breaking strength, breaking strain, breaking elongation, and modulus of elasticity of fiber samples. The pre-tension for the 1.46 dtex fiber is 0.1 cN, and for the 2.54 dtex fiber it is 0.2 cN. The test speed is 20 mm/min, and the measurement length is 10 mm.

## 3. Results and Discussion

### 3.1. Powder Particle Size analysis

Figure 3 shows the particle size distribution of the ZnO/TM composite powder. As shown in Figure 3, after mechanical ball milling, the average particle size of the composite powder was reduced to near 480 nm, with a particle distribution range of 150–1300 nm, which is obviously lower than the original tourmaline powder (average particle size of 1250 nm with a distribution range of 400–3600 nm). The reduction in functional powder particle size is conducive to increasing the specific surface area of the composite powder, improving the surface activity and reactivity of the powder [27]. This improvement directly affects the anion release performance of the ZnO/TM composite powder, aiding in achieving high-performance negative ion functional fibers. On the other hand, this reduction in particle size allows the powder to disperse in the PET substrate well. The interfacial bonding performance between the ZnO/TM composite particles and PET substrate could be enhanced, ensuring that the ZnO/TM/PET functional fibers can be evenly extruded from the spinneret and maintain excellent properties like tensile strength. Simultaneously, in textile industry standards, the usual rule for the particle size of the inorganic composite powder added to the fiber substrate is d50 ≤ 0.5 µm, d97 ≤ 3 µm. This provision ensures that the addition of inorganic powder does not significantly alter the fiber’s inherent properties and maintains a consistent distribution, thereby assisting the production of uniform and high-performance composite fibers. It can be seen that the newly developed composite powder can meet the preparation needs of new functional fibers from the perspective of particle size analysis.

### 3.2. Effect of Powder Content on Rheological Properties of Fiber Masterbatch

Figure 4 shows the apparent viscosity curves of the ZnO/TM/PET fiber masterbatch with shear rate after adding composite powders with different content at 270 °C. It can be seen from Figure 3 that compared with the rheological curve of pure PET slices, the apparent viscosity of fiber masterbatch decreases with the increase in compound powder addition. This shows that the addition of composite powder can reduce the viscosity of the system and enhance the masterbatch fluidity. For spinning, the masterbatch with better fluidity is more conducive to spinning processing and can increase the fiber yield. This may be due to the addition of composite powders affecting the regularity of polyester macromolecular chains, reducing the entanglement point of molecular chains, and the stacking structure of macromolecular chains becomes looser, resulting in an increase in the free volume of the matrix, which then results in a decrease in viscosity [28].

From Figure 4, it can be seen that the fluidity of the masterbatch is not significantly reduced due to the addition of high content ZnO/TM composite powder. This is because the ZnO/TM composite powder plays the role of “ball” lubrication in the polyester matrix. This phenomenon has a certain effect on promoting the spinnability of the functional masterbatch. During melt spinning, the shear rate of melt is in the range of 1000–10,000 s^−1^. It can also be seen from Figure 4 that the apparent viscosity of melt decreases rapidly within this range of shear rate, indicating that melt viscosity is very sensitive to shear rate. Therefore, in the process of masterbatch melt spinning, melt pressure and melt pump flow rate should be strictly controlled to ensure smooth spinning.

### 3.3. SEM Analysis

Figure 5 shows the microstructure and elemental analysis of the ZnO/TM composite powder. As can be seen from Figure 5a,b, compared with raw TM, the particle size of the ZnO/TM composite powder is significantly reduced, and the particle size distribution uniformity is significantly improved. The uniformity of this microstructure is essential for the production and preparation of fibers. It can be seen from Figure 5c that ZnO is effectively attached to the tourmaline surface during the ball milling process. The good stability of ZnO/TM composite powders ensures that they can maintain their chemical properties in the necessary external environment, which provides a stable “raw material” for subsequent fiber preparation. Secondly, the particle size distribution of the composite powder is stable, which helps to ensure the uniformity and consistency of the fiber in the textile process, thus improving the spinnability of the new functional fiber.

Figure 6 shows the micromorphology of ZnO/TM/PET fiber samples with different addition concentrations. As can be seen from Figure 6, the surface of the ZnO/TM/PET fiber is embedded with numerous ZnO/TM powder particles. With the increasing content of the ZnO/TM powder, the particle coverage on the fiber surface also increases, resulting in an increasingly rough surface of the fiber. The superfluous ZnO/TM particles are distributed on the fiber’s surface due to the physical adsorption. Furthermore, a small amount of agglomeration occurs, which leads to the blockage of spinneret and affects the spinnability of the fiber.

### 3.4. XRD

Figure 7 represents a comparison diagram of phase analysis of different samples. From Figure 7a, it can be observed that the main diffraction peaks of ZnO perfectly coincide with the standard diffraction peaks of ZnO in the database. The prepared ZnO/TM composite powder completely corresponds to the diffraction peaks of the ZnO standard card (PDF #36-1451) at 2θ = 31.769°, 34.421°, 36.252°, 47.538°, 56.602°, 62.862°, 66.378°, 67.961°, 69.098°, and 76.953°, and other diffraction peaks of the ZnO/TM composite powder mainly aligns with the main diffraction peaks of the raw TM. This indicates that in ZnO/TM composite powder, zinc oxide and tourmaline have successfully composited, and the fundamental structure of tourmaline has not been damaged, albeit with slightly lowered diffraction peaks than the original material.

Table 4 shows the crystallinity (%) and average particle size (nm) of ZnO/TM/PET fibers with different contents calculated according to Figure 7b. In Table 5, 010, 110, and 100 are specific crystal faces of pure polyester fibers, which represent different crystal structure directions or layers. The polyester chain is mainly arranged along the 010 direction, while the 100 and 110 sides are the faces perpendicular to the polyester chain arrangement direction. In Figure 7b, the diffraction peaks occurring at 17.5°, 22.4°, and 26.0° correspond to 010, 110, and 100 crystal faces, respectively. From Figure 7b and Table 4, it can be seen that with the increase in ZnO/TM composite powder content, the diffraction peaks near the three positions shift to the left, the peaks gradually weaken, and the crystallinity decreases. This is because the addition of a larger heteroatom than the main body makes the cell parameters larger and the crystal plane spacing larger, resulting in a leftward shift of the diffraction peak. At the same time, with the increase in ZnO/TM powder content, the crystalline zone of polyester fiber was destroyed more, which obstructs the arrangement and accumulation of polyester molecular chains and inhibits the growth of grain, forming more amorphous zones, which in turn reduces crystallinity. The tighter the structure is and the more ordered the crystallization zone is, the higher the durability and strength is [29]. With the increase in the ZnO/TM powder concentration, the amorphous zone of the fiber increases and the orientation of the fibers also decreases, resulting in a decrease in the side-by-side density of the fiber chains and an increase in the space between the molecular chains. As shown in Figure 7b, with the increase in ZnO/TM powder content, the diffraction peak of the fiber in the direction of 010 gradually weakens. The orientation of the fibers also decreases, resulting in a decrease in the side-by-side density of the fiber chains and an increase in the space between the molecular chains. This may reduce the mechanical properties such as tensile strength and elongation at break to some extent. 

### 3.5. FTIR

Figure 8 shows the FTIR spectrum of the fiber samples. As shown in Figure 8, it is evident that both types of fibers exhibit absorption peaks at 3438.8 cm^−1^, 1636.5 cm^−1^, 1404.8 cm^−1^, 1339.8 cm^−1^, 1239.2 cm^−1^, 1099.6 cm^−1^, 1014.2 cm^−1^, 871.7 cm^−1^, and 723.5 cm^−1^, corresponding, respectively, to the stretching vibrations of OH, bending vibrations of OH, bending vibrations of C-H, stretching vibrations of -CH_2_, stretching vibrations of C=O, stretching vibrations of C-O-C, bending vibrations of O-H, stretching vibrations of C-C, and out-of-plane stretching vibrations of the benzene ring’s C-H. Apart from the above absorption peaks, ZnO/TM/PET functional fibers exhibit characteristic absorption peaks of ZnO/TM powder at 1085.8 cm^−1^, 989.1 cm^−1^, 779.0 cm^−1^, 524.8 cm^−1^, 510.2 cm^−1^, 498.7 cm^−1^, and 429.8 cm^−1^. These peaks correspond to symmetric stretching vibrations of Si-O-Si, symmetric stretching vibrations of O-Si-O, symmetric stretching vibrations of Si-O-Si, bending vibrations of Zn-O, bending vibrations of Si-O, bending vibrations of BO_3_, and stretching vibrations of Zn-O. From the analysis of the above functional groups, it is clear that the ZnO/TM composite powder has successfully composited with the fiber substrate. This indicates that ZnO/TM powder has sufficient mixing and interaction with polyester matrix in the spinning process and produces stable bonding. The functional groups present in the ZnO/TM/PET composite negative ion functional fibers are similar to standard polyester fibers. But the presence of ZnO/TM may result in alterations in the chemical and physical properties of the fiber.

### 3.6. Effect of Powder Content on Mechanical Properties

Figure 9 shows the mechanical indexes of ZnO/TM/PET fibers with different fineness and content. It can be seen from Figure 9 that the mechanical properties of ZnO/TM/PET fibers display a decrease with the augmentation in the concentration of ZnO/TM powder. There could be two possible reasons for this. On the one hand, with the increase in the ZnO/TM powder concentration, a plethora of non-homogeneous distribution or micropores may form within the polyester fibers, potentially inducing internal stress, which further affects the mechanical properties of the fiber. On the other hand, excessive powder concentration might also impact the continuity of the polyester substrate, thereby compromising the overall toughness of the fiber, resulting in a decline in mechanical performance. From Figure 9, it can also be seen that the breaking strength and elastic modulus of 2.54 dtex fiber are lower than that of 1.46 dtex fiber, while the breaking elongation is higher than the latter. This is mainly related to the surface area of fiber and the internal microstructure of fiber. Compared with 1.46 dtex fibers, 2.54 dtex fibers have a larger volume but a smaller surface-to-volume ratio, resulting in a lower force transfer efficiency and thus a lower breaking strength. At the same time, a larger volume may have more internal micropores formed, through which greater deformation can be formed under the appropriate stress, resulting in an increase in the elongation at break and a decrease in the elastic modulus.

### 3.7. Thermal Performance Testing

Figure 10 is the DSC and TG curve of the samples. From Figure 10a, it can be seen that from 100 °C onwards, the DSC curve of raw TM shows a state of continuous heat release. This is related to the unstable structure of tourmaline. Since there is no continuously stable skeleton structure, changes in material structure lead to constant changes in DSC curves. The ZnO/TM composite powder has an obvious exothermic peak at about 700 °C, which indicates that the composite material containing zinc oxide can improve the interaction between elements in tourmaline minerals and provide higher stability. At temperatures greater than 700 °C, this improved stability may have led to some form of phase transition. However, the glass transition temperature of both is about 100 °C, and there is no significant change, which indicates that the short-range order degree of tourmaline and ZnO/TM composite powder is not significantly improved because of the preparation of composite powder. 

As can be seen from Figure 10b, both pure PET fiber and ZnO/TM/PET fiber have a large amount of heat release phenomenon at 253 °C, and the material structure changes. Compared with pure PET, the heat release peak of composite fiber is lower near 253 °C. This may be because the addition of ZnO/TM powder improves the thermal stability of the polyester and reduces the energy required. It can also be seen from Figure 10b that the glass transition temperature of the composite fiber is increased, reaching about 122.4 °C, and thermal stability is significantly improved.

From Figure 10c, it can be seen that the four stages of thermal decomposition of the two fiber samples were not significantly different. Compared with pure PET fiber, the initial decomposition temperature of the ZnO/TM/PET functional fiber is slightly lower, but the reduction degree is not obvious. This shows that the addition of composite powder has no great effect on the thermal stability of the matrix, and the fiber substrate still has good thermal stability, which can meet the high temperature requirements of subsequent spinning.

### 3.8. Thermal Shrinkage Performance Testing

Table 5 shows the thermal shrinkage ratio of pure PET fiber and ZnO/TM/PET functional fiber. It can be seen that the overall thermal shrinkage of composite fiber is lower than that of pure PET fiber, and the thermal shrinkage rate gradually decreases with the increase in added concentration. This is because the addition of ZnO/TM powder prevents the molecular chain from sliding due to external forces. So that the cooling and curing time of the fiber is extended, the molecular chain has enough time to reconfigure, thus reducing the thermal shrinkage rate of the fiber. As the concentration of the powder increases, the effect of preventing the molecular chain from sliding becomes more obvious, so the thermal shrinkage rate will be lower and lower. For the two fibers of different fineness, the thermal shrinkage of 2.56 dtex fiber is higher than that of 1.46 dtex fiber. This is because when the fiber is thinner, the degree of freedom of the internal chain segments of the molecular chain becomes smaller, and the spatial edge of the molecular chain configuration adjustment is narrow.

### 3.9. Negative Ion Release Performance Testing

Figure 11 illustrates the amount of NAI released by ZnO/TM/PET fibers with different concentrations of ZnO/TM power under both static and dynamic friction conditions. As demonstrated in Figure 11, with the increase in the concentration of ZnO/TM powder, the release of negative ions exhibits an increasing tendency. This is because with the increase in the powder content, more ZnO/TM powder is exposed to the air, and the resultant NAI concentration also increases. Furthermore, according to the test results, fibers of 1.46 dtex have a lower NAI release amount than fibers of 2.54 dtex. This is because fine fibers per unit mass have a larger surface area, exposing more ZnO/TM powder to the air and leading to increased NAI creation. Simultaneously, finer fibers may lead to a greater exposure of ZnO/TM powder, which increases the likelihood of contact with the air during friction.

## 4. Conclusions

A new ZnO/TM/PET fiber was prepared, and a series of physicochemical properties of this new fiber were analyzed. The results showed that the overall performance of the ZnO/TM composite is better than that of pure tourmaline powder and can meet the preparation needs of new functional fibers well. The addition of ZnO/TM composite powder has little effect on the fluidity of polyester masterbatch, which still shows shear thinning characteristics. ZnO/TM composite powder is randomly distributed on the surface and inside of ZnO/TM/PET functional fiber and has good dispersion. However, the addition of ZnO/TM powder reduces crystallinity, mechanical properties, and thermal shrinkage of the fibers. The negative oxygen ion release test shows that the increase in ZnO/TM powder concentration can promote NAI release under both static and dynamic conditions. So, ZnO/TM/PET fiber has good mechanical and thermal properties of textile fibers, can provide highly efficient air negative ions, and has great potential for industrial production.

## Figures and Tables

**Figure 1 polymers-16-01439-f001:**
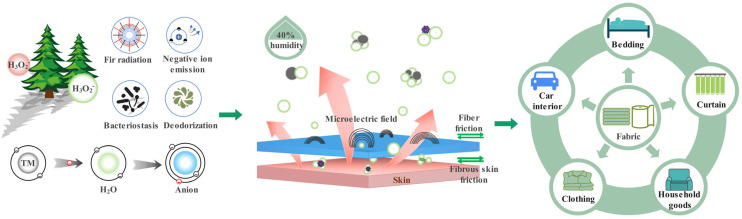
Application diagram of negative air ion functional fabric in industry and human life.

**Figure 2 polymers-16-01439-f002:**
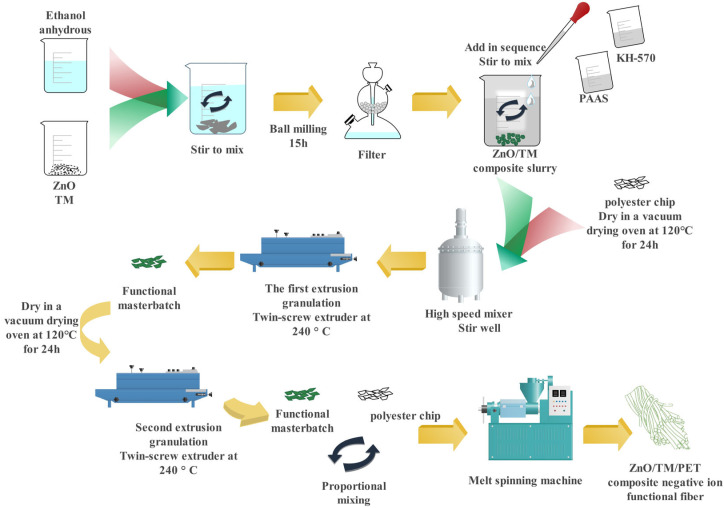
Process diagram of ZnO/TM/PET negative ion functional fiber.

**Figure 3 polymers-16-01439-f003:**
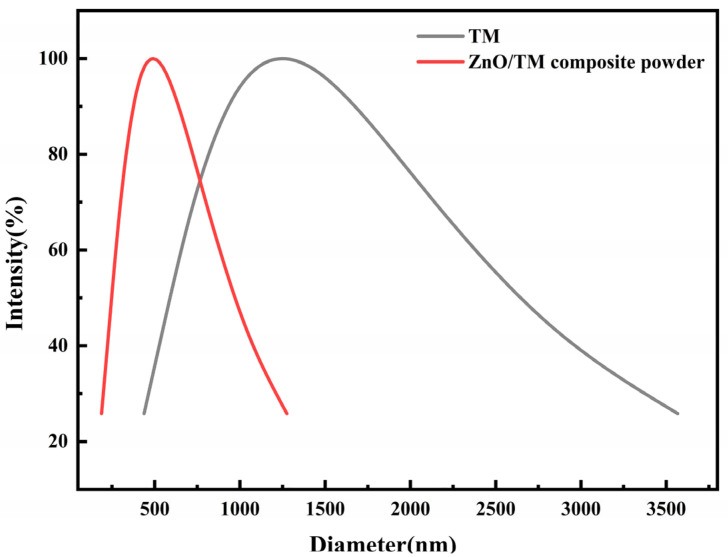
Particle size distribution of TM and ZnO/TM composite powder.

**Figure 4 polymers-16-01439-f004:**
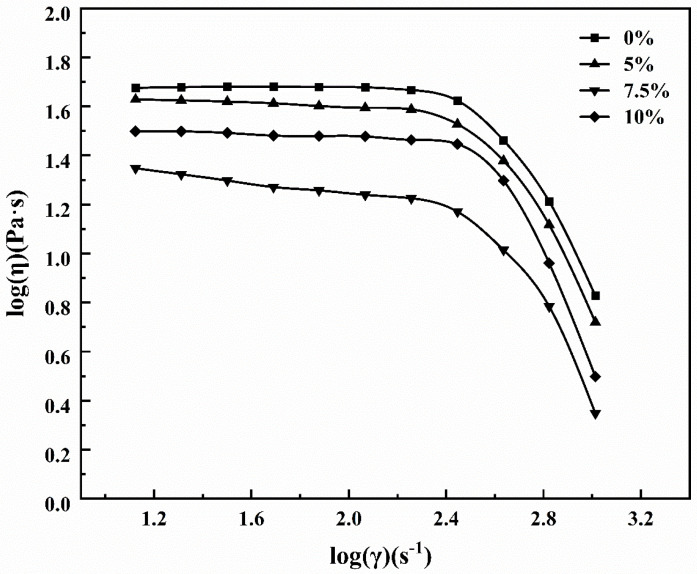
Rheological curves of ZnO/TM/PET functional masterbatch with different contents.

**Figure 5 polymers-16-01439-f005:**
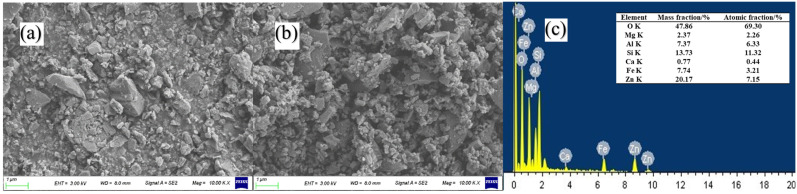
SEM and ESD of the sample: (**a**) raw tourmaline powder, (**b**) zinc oxide compound tourmaline powder, and (**c**) ESD atlas of the composite powder.

**Figure 6 polymers-16-01439-f006:**
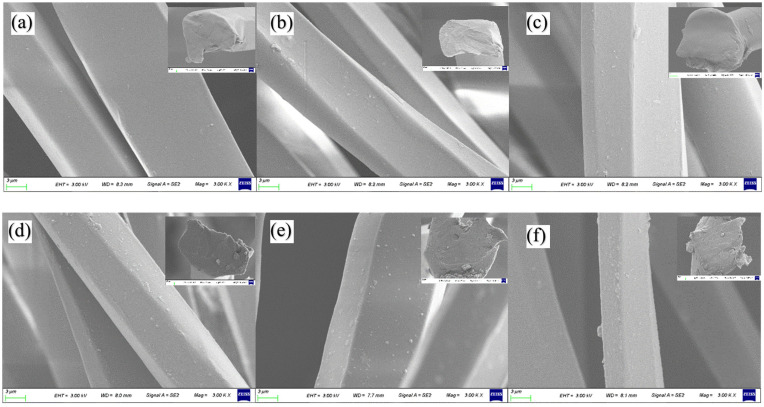
SEM photos of ZnO/TM/PET composite functional fibers: (**a**) 5%–2.54 dtex; (**b**) 5%–1.46 dtex; (**c**) 7.5%–2.54 dtex; (**d**) 7.5%–1.46 dtex; (**e**) 10%–2.54 dtex; (**f**) 10%–1.46 dtex.

**Figure 7 polymers-16-01439-f007:**
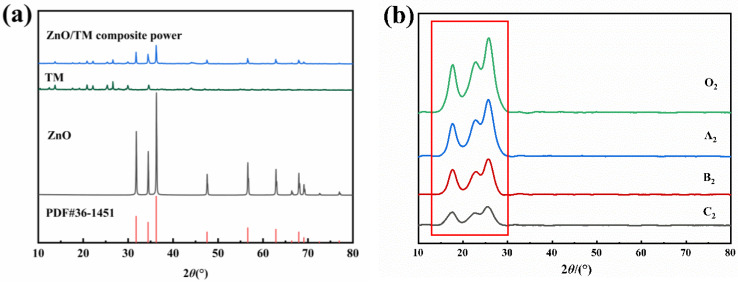
XRD patterns of the samples: (**a**) raw tourmaline and ZnO/TM composite; (**b**) ZnO/TM/PET fibers.

**Figure 8 polymers-16-01439-f008:**
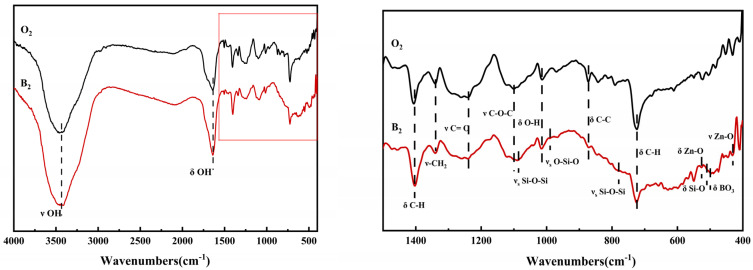
Infrared spectrum of a fiber sample.

**Figure 9 polymers-16-01439-f009:**
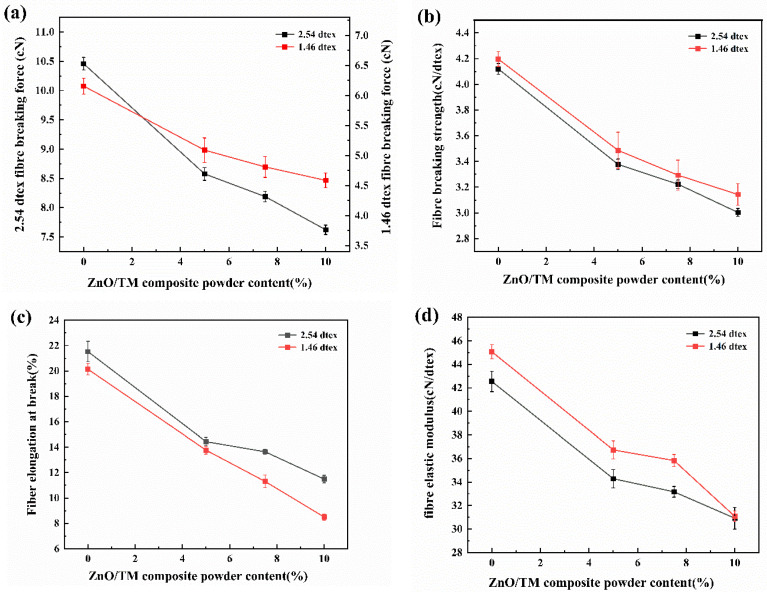
Mechanical properties curves of ZnO/TM/PET fiber samples: (**a**) breaking strength, (**b**) breaking strength, (**c**) elongation at break, and (**d**) elastic modulus.

**Figure 10 polymers-16-01439-f010:**
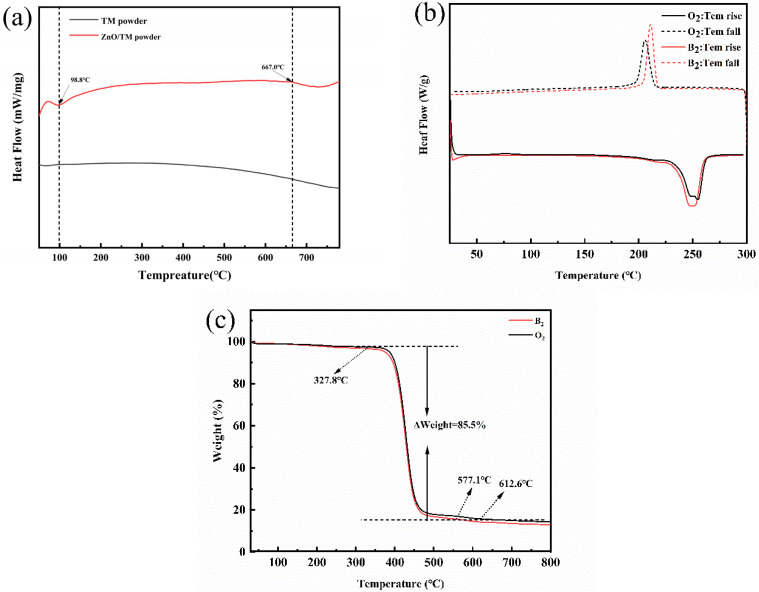
DSC and TG curve of the samples: (**a**) DSC curve of raw TM and ZnO/TM composite power; (**b**) DSC curve of PET and ZnO/TM/PET fibers; (**c**) TG curve of PET and ZnO/TM/PET fibers.

**Figure 11 polymers-16-01439-f011:**
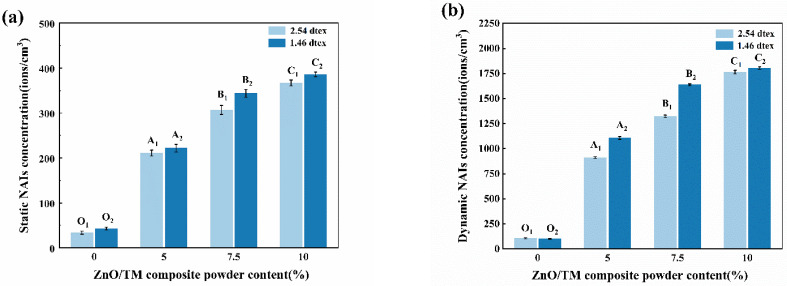
NAI release of different ZnO/TM/PET fiber samples: (**a**) static method, (**b**) dynamic method.

**Table 1 polymers-16-01439-t001:** Temperature of each zone of melt spinning equipment.

Process Condition	1st Area Screw Temp.	2nd Area Screw Temp.	3rd Area Screw Temp.	4th Area Screw Temp.	Screw Flange Temp.	Metering Pump Temp.	1st Area Hot Roll Temp.	2nd Area Hot Roll Temp.	3rd Area Hot Roll Temp.
SV (Actual temp) (°C)	202	260	270	270	269.9	265.2	65.1	66.3	68.2
PV (Set temp) (°C)	200	260	270	270	270.0	65.0	65.0	65.0	65.0

**Table 2 polymers-16-01439-t002:** Spinning process parameters.

Process Condition	Extruder Screw Frequency (Hz)	Extrusion Screw Current (A)	Metering Pump Speed (r/min)	Oiling Speed (r/min)	1st Area Hot Roll Speed (r/min)	2nd Area Hot Roll Speed (r/min)	3rd Area Hot Roll Speed (r/min)	Winding Speed (r/min)	Winding Angle (°)	Theoretical Drawing Ratio
1st Process parameter	2.00	8.90	5.0	3.0	300.0	450.0	470.0	500.0	5.0	1.7
2nd Process parameter	2.00	8.90	5.0	3.0	440.0	700.0	760.0	800.0	5.0	1.8

**Table 3 polymers-16-01439-t003:** Formulation and fineness of ZnO/TM/PET fibers.

Name	O_1_	O_2_	A_1_	A_2_	B_1_	B_2_	C_1_	C_2_
ZnO/TM composite powder (wt%)	0	0	5	5	7.5	7.5	10	10
PET (wt%)	100	100	95	95	92.5	92.5	90	90
Theoretical drawing ratio	1.7	1.8	1.7	1.8	1.7	1.8	1.7	1.8

**Table 4 polymers-16-01439-t004:** Crystallinity and average particle size of ZnO/TM/PET fiber samples.

Name	010		110		100		Crystallinity (%)	Average Particle Size D(nm)		
	2θ (°)	FWHM	2θ (°)	FWHM	2θ (°)	FWHM		010	110	100
O_2_	17.509	1.585	22.412	1.467	26.016	1.704	71.88	5.06	3.96	3.42
A_2_	17.572	1.641	22.452	1.458	25.961	1.725	69.01	5.04	3.96	3.43
B_2_	17.306	1.965	22.215	1.475	25.739	1.948	67.26	5.12	4.00	3.46
C_2_	17.286	2.014	22.155	1.418	25.771	1.968	65.27	5.13	4.01	3.45

**Table 5 polymers-16-01439-t005:** The thermal shrinkage ratio of PET and ZnO/TM/PET functional fiber.

Name	O1	O2	A1	A2	B1	B2	C1	C2
Fiber thermal shrinkage (%)	7.63	7.32	6.89	6.23	5.92	5.54	5.01	4.78

## Data Availability

The raw data supporting the conclusions of this article will be made available by the authors on request.

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
