# Peer review of "Preparation and Performance of a Novel ZnO/TM/PET Composite Negative Ion Functional Fiber"

_polymers, 2024, doi:10.3390/polym16101439_

Round 1
Reviewer 1 Report
Comments and Suggestions for Authors
The manuscript contains interesting studies on Novel ZnO/TM/PET Composite based on Negative Ion Functional Fiber However, the work requires a lot of corrections and additional information.
1. In the abstract part, from line 8-10, too lengthy sentence mentioned.please rewrite or modify the statement for clarification.
2. Need to add a graphical abstract for showing the air ion fabric applications in industry or other sectors for smooth understanding and their utilization.
3.In line 30-35 the sentence is too crowded and actual meaning is masked in so many words, please modify the sentence structure, and add 2- 3 references as well.
4. Ref 4 and 13 is repeated, please replaced
5. Line 57-66 please add more references from the literature
6. Sample preparation section Line 101, what was the rpm for grinding, and dimension of ball grinding tank?
7. In line 108-109 what was the ratio of each component?
8. In line 122- 131 the given data must be organized in tabulated form
8. In figure 1 the wording should be legible.
9. In line 167 and 176 what was the exact mass of fiber?
10. In Figure 3 please improve the overall quality of picture.
11. Line 247-250, is this a caption of figure 3 or a description. please modify it.
12. Figure 4 should replace with high quality picture.
13. All figures quality need to improve
14. Sentence structure in result and discussion section need to thoroughly improve.
15. Need to improve the conclusion part as well and must reflect the applicability of the fabricated fiber.
Comments on the Quality of English Language
I recommended to thoroughly improve the quality of English
Reviewer 2 Report
Comments and Suggestions for Authors
I examined thoroughly the manuscript Ref. No. polymers-2934397 entitled “Preparation and Performance of a Novel ZnO/TM/PET Composite Negative Ion Functional Fiber”.
Minor revision is required to improve the quality of the work. After that, the paper may be suitable for publication.
Full list of comments is appended in the following.
Comments
· Some typo and grammar errors throughout text should be carefully checked and corrected.
· Introduction section should be exhaustively revised and all recent literature should be included. The necessity, importance and novelty of the present work should be clearly highlighted and emphasized in the last part of “Introduction section.
· Full in detail specification for all material, methods and instruments should be given.
Following additional tests should be done and the results should be used in discussion on the results.
· Full in detail information about melt spinning condition? 1-Spinning temperature (extruder zones temperature); 2- cooling chamber temperature; 3- spinning speed;
· Rheological properties of ZnO/TM/PET composite (shear properties, storage modulus, loss modulus. etc) and its impact on the spinnability of composite should be well discussed.
· Term “Fracture” should be replaced with “breaking” in mechanical tests.
· Stress-strain curves of neat PET and composite PET fibers should be added?
· CV and SD of fibers?
· Thermal shrinkage of fibers should be assessed.
· DSC study of prepared composite and its prepared fiber should be done and discussed.
· Tg (glass transition temperature) of neat and composite PET fibers from both DSC and DMA tests should be assessed.
· Orientation of neat and composite PET fibers and its effects on mechanical properties should be studied.
· Crystallinity index, crystal size, etc of neat and composite PET fibers should be assessed and their impact of tensile/mechanical data should be discussed.
· Discussion should be made on the scaling up of the process into industrial applications.
· Discussion on the results should be extensively strengthened using up to date high quality recent literature in the field.
· Cost analysis of process should be done.
Comments on the Quality of English LanguageSome typo and grammar errors throughout text should be carefully checked and corrected.
